# Posterior Circulation Stroke Patients Receive Less Reperfusion Therapy Because of Late Arrival and Relative Contraindications: A Retrospective Study

**DOI:** 10.3390/jcm12165181

**Published:** 2023-08-09

**Authors:** Aleksandra Ekkert, Daiva Milmantienė, Unė Jokimaitytė, Dalius Jatužis

**Affiliations:** Faculty of Medicine, Vilnius University, LT-03225 Vilnius, Lithuania; daiva.milmantiene@mf.stud.vu.lt (D.M.); ujokimaityte@gmail.com (U.J.); dalius.jatuzis@mf.vu.lt (D.J.)

**Keywords:** posterior circulation stroke, reperfusion therapy, thrombolysis, thrombectomy, relative contraindications, late arrival, FAST, BEFAST, stroke recognition, decision-making

## Abstract

Background. Reperfusion treatment (RT) is administered to individuals with posterior circulation strokes (PCS) later and less frequently. We aimed to study the impact of demographic and clinical factors on the decision for RT in PCS. Methods. We conducted a retrospective analysis of the data from 500 subjects admitted to the tertiary stroke centre’s emergency department between 2018 and 2020 due to PCS. Demographic and clinical factors were analysed among three groups: the RT group, the group with no RT because of absolute contraindications (ACI), and the group with no RT because of relative contraindications (RCI). Results. Of the patients, 202 (40.3%) were female. The median NIHSS was four (4), and the subjects’ median age was 69 (18). RT was performed on 120 (24%) subjects. FAST symptoms (OR—5.62, 95% CI [2.90–12.28]) and higher NIHSS (OR—1.13, 95% CI [1.09–1.18]) at presentation, atrial fibrillation (OR—1.56, 95% CI [1.02–2.38]), hypertension (OR—2.19, 95% CI [1.17–4.53]) and diabetes (OR—1.70, 95% CI [1.06–2.71]) increased the chance of RT. Late arrival was the most prevalent ACI for 291 (58.2%) patients. FAST-negative subjects (OR—2.92, 95% CI [1.84–4.77]) and males (OR—1.58, 95% CI [1.11–2.28]) had a higher risk of arriving late. Because of RCI, 50 (10%) subjects did not receive RT; the majority were above 80 and had NIHSS ≤ 5. Subjects with RCI who received the RT had a higher NIHSS (4 vs. 3, *p* < 0.001), higher hypertension (59 (92.2%) vs. 35 (77.8%), *p* = 0.032) and heart failure (23 (35.9%) vs. 7 (15.6%), *p* = 0.018) prevalence. There was a trend for less RT in females with RCI. Conclusions. Late arrival was the most common barrier to RT, and the male gender increased this risk. because of relative contraindications, 10% of subjects were not considered for RT. The presence of FAST symptoms, vascular risk factors, and a higher NIHSS increased the chance of RT.

## 1. Introduction

Posterior circulation strokes (PCS) occur in approximately 20% of all ischemic strokes [1]. However, the reported percentage of PCS among the patients treated with reperfusion therapy (RT) is lower—5–19% [2]. Specific anatomic and hemodynamic properties of the posterior circulation, such as lower flow velocities, different vessel calibre, and even different clot structures, result in distinct stroke aetiology and clinical course, compared to the anterior circulation [3,4].

PCS presents with typical symptoms listed in the FAST stroke recognition tool (face asymmetry, arm weakness, speech disturbance) less frequently. FAST tends to miss 40% of PCS, while the BEFAST tool, comprising balance and eye symptoms, is more sensitive [5,6]. PCS manifest with non-typical symptoms, like vomiting and seizures [7], more frequently compared to anterior circulation strokes (ACS). Patients with PCS are at risk of belated arrival at the hospital [8], and thrombolysis rates are lower in this group [9]. PCS patients are managed slower and receive RT later than those with ACS [10,11].

Not all the PCS symptoms are included in the National Institutes of Health Stroke Scale (NIHSS), sometimes resulting in a hesitancy to apply RT in PCS. Due to the non-typical presentation and relatively low NIHSS scores, weighting RT effectiveness and safety is quite complicated in some cases. Although the risk of symptomatic intracerebral haemorrhage in PCS is lower than in ACS, the dilemma of RT risk and benefit always exists, especially when NIHSS scores are low [12,13]. In such cases, decision-making might be guided by subtle, often subjective factors, not covered by the guidelines.

In light of these findings, it seems reasonable to identify the reasons for the scarcity of RT in PCS patients and the factors contributing to the decision to withhold RT. In this study, we aimed to analyse demographic and clinical factors influencing the decision for RT in PCS.

## 2. Materials and Methods

This retrospective observational single-centre study was conducted at Vilnius University Hospital Santaros Klinikos—a comprehensive stroke centre with a catchment population of 945,000. The research population included 500 subjects admitted due to ischemic PCS from January 2018 to December 2020. We did not continue further recruitment of the patients during the COVID-19 pandemic because it could result in longer times for treatment and arrival. This would make the data inappropriate for use after the pandemic.

The inclusion criteria were the following:Ischemic PCS diagnosis. PCS diagnosis was confirmed in every subject either using neuroimaging (ischemia on the plain computerized tomography (CT) scan, posterior circulation vessel occlusion on the CT angiography, hypoperfusion in the posterior circulation territory on the CT perfusion or diffusion-weighted imaging-positive lesion in the posterior circulation, all of these needed to correspond to clinical symptoms), or based on typical clinical symptoms (e.g., alternating brainstem syndrome).Aged 18 years old or older. There was no upper age limit for inclusion.Hospitalised at the same centre.

The exclusion criteria were the following:Transfer to other hospitals after the initial evaluation in the emergency department.Unclear stroke territory.Both ACS and PCS were detected.

We analysed clinical and demographic factors that influenced the suitability for the RT, including absolute and relative contraindications (ACI and RCI, respectively), according to the hospital protocol (Table 1). 

Subjects treated with intravenous thrombolysis (IVT), mechanical thrombectomy (MT) or both methods (bridging therapy (BT) were included in the RT group, and subjects who did not receive any RT were included in the non-RT group. The non-RT group was further divided into two subgroups: subjects who had ACI according to the hospital reperfusion treatment protocol (ACI group), and subjects who did not receive RT despite not having any ACI. This group did not receive RT due to factors that were not established as clear ACI in the hospital protocol, and the group was labelled as the relative contraindications group (RCI group). RCI included minor stroke (NIHSS ≤ 5 points), age > 80 years, ischemic stroke history within the past 3 months, intracranial aneurysm, and presenting with a seizure. The cases of posterior cerebral artery occlusion were discussed with the interventional radiologists who performed MT. Some of those cases were reported as “technically risky interventions”. If MT was not performed due to this, the case was classified as a relative contraindication. The criteria used for the inclusion of subjects into the particular subgroup are presented in Table 2.

If the subject had several ACI, only the most important, determinant contraindication was registered (e.g., if a subject was late and had established infarct on plain CT, such a case was classified as belated arrival). In the RCI group, if the subject had several RCIs, all of them were registered. 

Some subjects who had RCI received RT. We compared the baseline characteristics and outcomes between the subjects with RCI who received the RT and those who did not. Outcomes compared were lethal outcome during the hospitalization (labelled as an early lethal outcome), early ambulatory outcome defined as mRS 0–3 points (able to walk) on discharge from the stroke centre, and in-patient complications: intracranial or another major bleeding, recurrent stroke, myocardial infarction, infection, delirium.

Baseline characteristics included demographic data (sex, age), clinical symptoms at presentation, baseline NIHSS score, medical history (arterial hypertension (AH), congestive heart failure NYHA B or worse (CHF), history of stroke or transient ischemic attack (TIA), myocardial infarction (MI), diabetes mellitus (DM), and atrial fibrillation (AF).

All patients were examined by a neurology consultant on admission. The decision for RT was based on clinical and imaging findings. IVT was performed within 4.5 h after symptom onset and 6 h in the case of basilar artery occlusion (BAO). For IVT, a 0.9 mg/kg dose of alteplase was used (a maximum dose of 90 mg), with 10% of the dose given as a bolus in 1–2 min and the rest given as an intravenous infusion for 1 h. Mechanical thrombectomy was performed within 6 h of onset and within 24 h in the case of BAO. If the subject arrived later than the recommended timeframe for the RT, the case was classified as a late arrival.

Data were analysed using R software version 4.2.1. (R Core Team (2022)). Baseline characteristics are reported using descriptive statistics. The normality was assessed using the Shapiro-Wilk test; all qualitative variables were not normally distributed. The chi-square test was used to compare qualitative variables between groups, and the Wilcoxon test was used to compare quantitative variables between two groups. The accepted level of statistical significance was <0.05. Univariate logistic regression was used to analyse the odds ratios with a 95% confidence interval. The study power was 0.879 (calculated with G-Power software, version 3.1.9.2.).

The study was approved by the Vilnius Regional Bioethics Committee (approval Nr. 1170, 19 December 2019) and the Lithuania Bioethics Committee (approval Nr. L-14-03/1, L-14-03/2, L-14-03/3, L-14-03/4, L-14-03/5, L-14-03/6, 18 April 2014).

## 3. Results

### 3.1. Baseline Characteristics

The median age was 69 (18) years, the median NIHSS on admission was four (4) points, and 202 (40.3%) subjects were female. The number of FAST-positive (FAST+) subjects was 372 (74.4%). Other subjects were FAST-negative: 53 (10.6%) presented with ataxia, 23 (4.6%) with vision disturbance only (visual field deficit or double vision), and 39 (7.8%) presented with both ataxia and eye symptoms, resulting in 487 (97.4%) BEFAST-positive (BEFAST+) subjects in total. The most frequent symptom at presentation was ataxia (63.6%), followed by paresis (54.2%) and speech disturbance (51.8%). The prevalence of all symptoms is listed in Figure 1.

RT was performed on 120 (24%) subjects: 72 (14.4%) were treated with IVT, 37 (7.4%) with MT, and 11 (2.2%)—with BT. Subjects in the RT group had higher NIHSS (7 vs. 3, *p* < 0.001), and more of them were FAST + (92.6% vs. 68.8%, *p* < 0.001) and BEFAST+ (100% vs. 96.6%, *p* = 0.039). In the RT group, there was a higher frequency of AF (42.1% vs. 32.2%, *p* = 0.045), AH (90.9% vs. 81.8%), and DM (29.8% vs. 19.3%) (Table 3).

### 3.2. Contraindications

330 (66%) subjects had ACI to RT. The most frequent ACI was late arrival: 291 (58.2%) subjects arrived later than the recommended timeframe for the appropriate RT. In 50 (10%) subjects, RT was not applied due to RCI only. Age > 80 years and NIHSS ≤ 5 points were the most frequent reasons to withhold RT. In this group, 15 (30%) subjects had more than one RCI. Of the subject with RCI, 49 (98%) were BEFAST+. Two (4%) of them did not have a disabling deficit (only isolated dizziness or isolated mild speech disturbance). The detailed structure of contraindications is presented in Table 4.

Having balance or visual symptoms without FAST symptoms was associated with an almost three-fold increase in the risk of late arrival (OR—2.92, 95% CI [1.84–4.77]). Male sex was another significant risk factor (OR—1.58, 95% CI [1.11–2.28]). Factors decreasing the risk of late arrival were the presence of FAST symptoms (OR—0.31, 95% CI [0.19–0.49]), a higher NIHSS score (OR—0.88 for each point, 95% CI [0.85–0.92]), AF (OR—0.51, 95% CI [0.35–0.74]) and heart failure (OR—0.51, 95% CI [0.34–0.76]) (Table 5). The last two may not have been independent risk factors confounded by the NIHSS, as they were higher in AF and HF subjects (five in AF and HF subjects vs. three in non-AF and non-HF subjects, *p* < 0.001).

### 3.3. Reperfusion Therapy

Being FAST+ was the most significant factor for receiving RT (OR—5.62, 95% CI [2.90–12.28]). Other factors increasing the chance of receiving RT were higher NIHSS (OR—1.13 for each point, 95% CI [1.09–1.18]), history of AF (OR—1.56, 95% CI [1.02–2.38]), AH (OR—2.19, 95% CI [1.17–4.53]) and DM (OR—1.70, 95% CI [1.06–2.71]) (Table 6).

Although some subjects did not receive RT due to RCI, such as age and minor stroke, others with the same RCI were treated with RT. To clarify whether there were any additional factors influencing the decision, we compared baseline characteristics between RT and RCI groups, including only the subjects who had the most common RCI, i.e., being 80 years of age or older and having a minor stroke, defined as NIHSS ≤ 5. There were 64 subjects with the aforementioned RCI in the RT group. They had significantly higher NIHSS (four vs. three, *p* < 0.001) and higher prevalence of AH (92.2% vs. 77.8%, *p* = 0.032) and HF (35.9% vs. 15.6%, *p* = 0.018) than the RCI group. There was a trend for lower female prevalence in the RT group with RCI (39.1% vs. 57.8%, *p* = 0.054). None of the outcomes, including early in-hospital mortality, early ambulatory outcomes, or complication rates, differed between the groups (Table 7).

## 4. Discussion

### 4.1. Discussion

The most frequent obstacle to receiving RT was belated arrival in more than half of the subjects. This finding confirms the results of other studies that revealed that PCS is a risk factor for late arrival and not receiving RT [8]. More PCS subjects arrived in time and received RT when they were FAST-positive. These findings can reflect two aspects.

First of all, after numerous educational campaigns, stroke symptoms listed in the FAST test might be better recognized by society and clinicians [15,16,17]. Nevertheless, about 20% of PCS subjects did not have FAST symptoms but had balance impairment, vision disturbance, or both of them. An Austrian study of PCS patients shows that PCS is associated with significant delays in prehospital and intra-hospital management. These findings show that there is room for improvement by initiating further educational campaigns, now using the BEFAST tool.

Furthermore, in those patients who arrived timely, the clinician‘s decision might have been biased, as ataxia and visual symptoms are sometimes underscored by the NIHSS [18]. Such strokes might be classified as minor strokes, causing doubt and incertitude about the risks and benefits of RT. Nevertheless, ataxia and vision disturbance are disabling symptoms, and, ideally, modified NIHSS scores for posterior circulation should be used to estimate potential disability [19,20,21]. Moreover, recent data highlight the effectiveness of RT even on such subtle outcomes as vision and cognitive functions [22].

Ten percent of subjects were not recognized as suitable for RT with no strict contraindications. However, RT happened in a large proportion of subjects with the same RCI, and, again, the most significant factor that differed between the groups who received RT and did not was the median NIHSS score (four vs. three). Nevertheless, many so-called “minor strokes” were present in both groups. One can argue that only the subjects with non-disabling deficits were not thrombolysed, but only 13 subjects from the whole cohort were BEFAST-negative, and nobody from the BEFAST-positive cohort presented with isolated facial asymmetry, which means that they all had disabling symptoms. These findings clearly show the subjective component of the assessment that is always present on top of guidelines.

Thrombolysis in minor strokes is a constant matter of debate between stroke physicians. Some authors report that RT does not have a beneficial impact on the outcomes of minor strokes [23], while others are more optimistic [24], especially when large-vessel occlusion is present [25]. Unfortunately, the outcomes measured are usually mortality, mRS and NIHSS, and those are quite crude. To draw a reliable conclusion about minor stroke outcomes, it would be beneficial to investigate other aspects, including cognitive functions, fatigue, autonomic dysfunction [26] and other often underestimated stroke symptoms.

We can hypothesise that RCI “age >80” could be a vague description of the subjective clinician’s impression of the subject’s frailty. It is known that frailty increases the probability of poor outcomes [27,28,29], and sometimes it is hard to predict if the frail patient will benefit from RT. Although RT in the elderly is another questionable concept among clinicians, age should not be a contraindication for RT per se [30].

Subjects diagnosed with AF, AH, and DM were treated with RT more frequently. While AF might not be an independent risk factor because of the confounding with the NIHSS, AH and DM were not associated with higher NIHSS scores. We hypothesise that the presence of vascular risk factors made subjects and clinicians more vigilant about stroke. In our opinion, RT could have been withheld in the RCI group because of the uncertainty about the diagnosis. In such cases, additional cardiovascular risk factors encouraged the clinician to suspect stroke and make the choice in favour of RT.

Although males were late for RT more frequently, there was a trend to withhold RT in females with RCI for RT. That finding reconfirms the results of previous studies regarding sex differences in stroke RT, showing that women are receiving less RT than men, even after adjusting for confounders [31,32,33]. It is another illustration of the subjectivity of the decision-making process.

A study from Portugal identified that social factors such as poverty, lack of stroke awareness, or difficulties in requesting immediate medical help put patients at higher risk of late admission for RT [8]. Given this, we should aim to increase awareness of PCS symptoms by educating society about stroke symptoms with the help of the BEFAST tool. The data from the tertiary centre in Helsinki revealed that PCS patients have hypertension history less often and presented with non-typical symptoms, such as seizure, vomiting and headache more often than the ACS patients [7]. We encourage clinicians to consider the possibility of stroke even when the patient does not have traditional cardiovascular risk factors or presents with atypical symptoms. Testing for specific PCS symptoms that are not represented by classical NIHSS, such as axial ataxia and dysphagia, might be helpful. It is beneficial to remember that thrombolysis in stroke mimics is safe [34,35] and that IVT is recommended in minor strokes if the deficit is disabling [30]. In our opinion, it would be beneficial to analyse the impact of routine use of the BEFAST tool by paramedics and in the Accident and Emergency Department on large cohorts in future.

### 4.2. Strengths and Limitations of the Study

Our study is limited by its retrospective design. Moreover, some of the PCS patients transferred to other centres were not included because the exact diagnosis and follow-up would be complicated. However, it does not make our findings less relevant. All transferred subjects were not suitable for the RT, so the real number of subjects with contraindications could be even higher. We did not analyse the impact of smoking status and dyslipidaemia on the chance of receiving RT because data about these factors were lacking. We also did not stratify our population according to stroke aetiology or type of admission (self-presented or paramedics). These problems should be investigated in future studies.

The strengths of the study are sufficient sample size and reliable clinical examination; every subject was examined by a neurology consultant on admission. It is also important that the majority of subjects had a radiologically confirmed PCS diagnosis.

## 5. Conclusions

Late arrival was the most common ACI to RT, and the male gender increased this risk. PCS patients with existing FAST symptoms, vascular risk factors, and higher NIHSS scores were more likely to be selected for reperfusion therapy. Ataxia or visual symptoms reduced the chance of receiving RT. Ten percent of subjects did not undergo RT due to relative contraindications. When only relative ineligibility criteria were present, RT was more often performed in the presence of higher NIHSS scores and vascular risk factors. There was a trend towards less frequent RT in female patients with relative contraindications. 

## Figures and Tables

**Figure 1 jcm-12-05181-f001:**
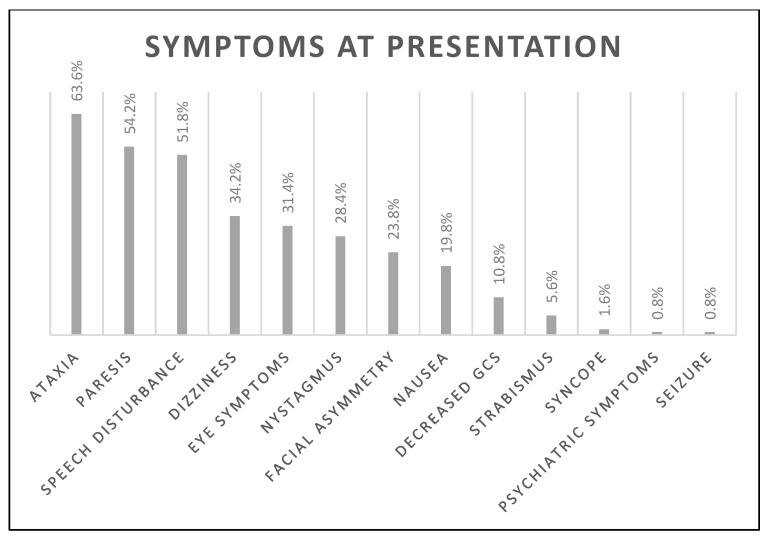
The prevalence of all symptoms on admission is presented in the figure. Eye symptoms include any visual disturbance: amblyopia, hemianopia, scotoma, diplopia. Decreased GCS—decreased level of consciousness measured using the Glasgow Coma Scale.

**Table 1 jcm-12-05181-t001:** Absolute contraindications for the reperfusion therapy in the study.

Absolute Contraindication	Study Label
Suitable for intravenous thrombolysis (IVT) only, treatment cannot be applied within 4.5 h (6 h in the case of basilar artery occlusion)	Late arrival
Suitable for mechanical thrombectomy (MT) only, treatment cannot be applied within 24 h	
Using warfarin, INR ≥ 1.7	Anticoagulant use
Direct oral anticoagulants used in less than 48 h	
Low-molecular-weight heparin used in less than 12 h	
Heparin use with APTT two times higher than the upper normal range limit and impossible to reverse in time	
mRS > 2 points	mRS > 2
Established stroke occupying more than 1/3 of the middle cerebral artery territory on the plain head CT	Established ischemia
Unfavourable penumbra-core ratio according to the ESO-ESMINT guidelines [14]	
Major bleeding within the past 3 weeks	Recent major bleeding
Major surgery within the past 3 weeks	Recent major surgery
ICH history	ICH history

This table represents absolute contraindications according to the hospital protocol and the labels used to classify them in the study. APTT—activated partial thromboplastin time, CT—computerised tomography, ESMINT—European Society for Minimally Invasive Neurological Therapy, ESO—European Stroke Organisation, ICH—intracerebral haemorrhage, INR—international normalised ratio, IVT—intravenous thrombolysis, mRS—Modified Rankin scale, MT—mechanical thrombectomy.

**Table 2 jcm-12-05181-t002:** Subgroup classification criteria.

RT Group: Any RT Method Applied	Non-RT Group: No RT Applied	
	ACI Subgroup	RCI Subgroup
Subjects who were treated with: IVT MT IVT + MT (BT)	Subjects who were not treated with RT due to ACI, according to the hospital protocol: Arrived too late to be treated with RT. Any anticoagulant use preventing the patient from receiving RT. mRS > 2 points. Established stroke occupying more than 1/3 of the middle cerebral artery territory on the plain head CT or unfavourable penumbra-core ratio according to the ESO-ESMINT guidelines. Major bleeding or surgery within the past 3 weeks. History of ICH.	Subjects who were not treated with the RT in the absence of the ACI, according to the hospital protocol, such as: NIHSS ≤ 5 points. Age > 80 years. Ischemic stroke history within the past 3 months. Unruptured intracranial aneurysm. Presenting with a seizure.

**Table 3 jcm-12-05181-t003:** Comparison of baseline characteristics in subjects who were and were not treated with reperfusion therapy.

Factor	RT Group	Non-RT Group	*p*-Value
Atrial fibrillation	51 (42.1%)	122 (32.2%)	0.045 *
Age	69 (15)	69 (19)	0.528
Hypertension	110 (90.9%)	310 (81.8%)	0.017 *
Anticoagulant use	14 (11.6%)	65 (17.2%)	0.143
Antiplatelet use	19 (15.7%)	70 (18.5%)	0.488
BEFAST+	120 (100%)	366 (96.6%)	0.039 *
Diabetes	36 (29.8%)	73 (19.3%)	0.015 *
Female sex	49 (40.5%)	153 (40.4%)	0.980
FAST+	112 (92.6%)	260 (68.6%)	<0.001 *
Heart failure	37 (30.6%)	90 (23.7%)	0.133
History of stroke or transient ischaemic attack	19 (15.7%)	85 (22.4%)	0.113
History of myocardial infarction	24 (19.8%)	78 (20.6%)	0.859
NIHSS	7 (7)	3 (3)	<0.001 *
Ongoing malignancy	4 (3.3%)	26 (6.9%)	0.152

Quantitative and numeric ordinal data (age in years and NIHSS in points) are presented as the median and interquartile range (IQR). Qualitative data are presented as an absolute number (percentage). * Significant differences are denoted with an asterisk. BEFAST+—at least one of the following symptoms: loss of balance, visual or eye disturbance, face asymmetry, arm weakness, speech disturbance; FAST+—at least one of the following symptoms: face asymmetry, arm weakness, speech disturbance; NIHSS—National Institutes of Health Stroke Scale, RT—reperfusion therapy.

**Table 4 jcm-12-05181-t004:** Contraindications for RT in PCS subjects.

ACI		RCI	
Belated arrival	291 (58.2%)	NIHSS ≤ 5 points	38 (9.6%)
Anticoagulant use	22 (4.4%)	Age > 80 years	19 (4.8%)
mRS > 2 points	9 (1.8%)	High subjective risk of MT	6 (1.2%)
Recent major bleeding	3 (0.6%)	Stroke in 3 months	1 (0.2%)
Established infarct on plain CT	3 (0.6%)	Intracranial aneurysm	1 (0.2%)
Recent major surgery	1 (0.2%)	Seizure	1 (0.2%)
ICH history	1 (0.2%)	More than 1 RCI	15 (3.0%)

Absolute and relative contraindications for reperfusion therapy are listed in the table. The percentage of the total study population is denoted in round brackets. ACI—absolute contraindications, CT—computerized tomography, MT—mechanical thrombectomy; ICH—intracerebral haemorrhage, mRS—Modified Rankin scale, NIHSS—National Institutes of Health Stroke Scale, PCS—posterior circulation stroke, RCI—relative contraindications, RT—reperfusion therapy.

**Table 5 jcm-12-05181-t005:** Association of demographic and clinical factors with the risk of belated arrival.

Factor	B	Std. Error	*p*-Value	OR (Exp B)	95% CI
AF	−0.68	0.19	<0.001 *	0.51	0.35–0.74 *
Age ≥ 80 years	−0.24	0.21	0.248	0.78	0.51–1.19
AH	−0.47	0.26	0.067	0.62	0.37–1.02
Balance or vision disturbance	1.07	0.24	<0.001 *	2.92	1.84–4.77 *
DM	−0.16	0.22	0.451	0.85	0.55–1.30
FAST+	−1.16	0.24	<0.001 *	0.31	0.19–0.49 *
HF	−0.68	0.21	0.001 *	0.51	0.34–0.76 *
History of stroke or TIA	−0.13	0.22	0.572	0.88	0.57–1.37
Male sex	0.46	0.18	0.012 *	1.58	1.11–2.28 *
NIHSS (risk reduction for each additional point)	−0.12	0.02	<0.001 *	0.88	0.85–0.92 *

Data are presented as odds ratio and 95% confidence interval. * Significant differences are denoted with an asterisk. AF—atrial fibrillation, AH—arterial hypertension, CI—confidence interval, DM—diabetes mellitus, FAST+—at least one of the following symptoms: face asymmetry, arm weakness, speech disturbance; HF—heart failure, NIHSS—National Institutes of Health Stroke Scale, OR—odds ratio, Std. Error—Standard Error; TIA—transient ischemic attack.

**Table 6 jcm-12-05181-t006:** Association of demographic and clinical factors with the chance of receiving RT.

Factor	B	Std. Error	*p*-Value	OR (Exp B)	95% CI
AF	−0.84	0.61	0.171	1.56	1.02–2.38 *
Age ≥ 80 years	−0.22	0.26	0.402	0.81	0.48–1.32
AH	0.79	0.34	0.022 *	2.19	1.17–4.53 *
Balance or vision disturbance	−1.56	0.37	<0.001 *	0.21	0.10–0.41 *
DM	0.53	0.24	0.026 *	1.70	1.06–2.71 *
FAST+	1.73	0.36	<0.001 *	5.62	2.90–12.28 *
HF	0.36	0.23	0.118	1.44	0.91–2.25
History of stroke or TIA	−0.43	0.28	0.126	0.65	0.37–1.11
Male sex	−0.02	0.21	0.912	0.98	0.64–1.49
NIHSS (chance increase for each additional point)	0.12	0.02	<0.001 *	1.13	1.09–1.18 *

Data are presented as odds ratio and 95% confidence interval. * Significant differences are denoted with an asterisk. AF—atrial fibrillation, AH—arterial hypertension, CI—confidence interval, DM—diabetes mellitus, FAST+—at least one of the following symptoms: face asymmetry, arm weakness, speech disturbance; HF—heart failure, MI—myocardial infarction, NIHSS—National Institutes of Health Stroke Scale, OR—odds ratio, RT—reperfusion therapy, TIA—transient ischemic attack.

**Table 7 jcm-12-05181-t007:** Baseline characteristics and outcomes in subjects with relative contraindications, compared between the RT and RCI groups.

Factor/Outcome	RT Group (N = 64)	RCI Group (N = 45)	*p*-Value
AF	29 (45.3%)	16 (35.6%)	0.308
AH	59 (92.2%)	35 (77.8%)	0.032 *
Antiplatelet use	13 (20.3%)	12 (26.7%)	0.437
BEFAST+	64 (100%)	44 (97.8%)	0.231
DM	16 (25%)	8 (17.8%)	0.370
FAST+	55 (85.9%)	34 (75.6%)	0.168
HF	23 (35.9%)	7 (15.6%)	0.018 *
History of MI	15 (23.4%)	8 (17.8%)	0.476
Ongoing malignancy	3 (4.7%)	1 (2.2%)	0.500
Female sex	25 (39.1%)	26 (57.8%)	0.054 *
History of stroke or TIA	9 (14.1%)	9 (20%)	0.411
Age	76 (19%)	76 (19%)	0.751
NIHSS	4 (3)	3 (3)	<0.001 *
Early ambulatory outcome	26 (40.6%)	19 (42.2%)	0.868
Delirium	7 (10.9%)	4 (8.9%)	0.727
Intracranial haemorrhage	0 (0%)	0 (0%)	0.068
Another bleeding	0 (0%)	1 (2.2%)	0.231
Myocardial infarction	2 (3.1%)	1 (2.2%)	0.777
Infection	17 (26.6%)	9 (20%)	0.429
Lethal outcome	3 (4.7%)	0 (0%)	0.141
Recurrent stroke	3 (4.7%)	0 (0%)	0.141

Quantitative and numeric ordinal data are presented as median (IQR). Qualitative data are presented as an absolute number (percentage). * Significant differences are denoted with an asterisk. AF—atrial fibrillation, AH—arterial hypertension, BEFAST+—at least one of the following symptoms: loss of balance, visual or eye disturbance, face asymmetry, arm weakness, speech disturbance; DM—diabetes mellitus, FAST+—at least one of the following symptoms: face asymmetry, arm weakness, speech disturbance; HF—heart failure, MI—myocardial infarction, NIHSS—National Institutes of Health Stroke Scale, RCI—relative contraindications, RT—reperfusion therapy, TIA—transient ischemic attack.

## Data Availability

Data is available on request due to privacy restrictions.

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
