# Peer review of "Posterior Circulation Stroke Patients Receive Less Reperfusion Therapy Because of Late Arrival and Relative Contraindications: A Retrospective Study"

_jcm, 2023, doi:10.3390/jcm12165181_

Round 1

Reviewer 1 Report

The problem of the timing of re-perfusion therapy in ischemic stroke has been actively studied for many years, but there are a number of open questions to date. The relevance of this study is beyond doubt.

However, the manuscript needs a major revision.

The stylistics of the English language, starting from the abstract and further, should be improved by the authors. I recommend attracting a native English speaker.

Abstract.

Please note that the purpose here and in the Introduction section should be identical.

Don't start sentences with numbers.

Add the mean age of the patients.

Add the missing spaces between the digits (for example, lines 13, 14, 15 and beyond).

 Materials and methods.

Please add a description of the type of this study.

Add information about the approval of the study by the Ethics Committee, including the institution, registration number and the date of the protocol.

Formulate clear criteria for inclusion and exclusion of patients in this study.

Add criteria for forming three groups of participants, including their minimum and maximum age.

How was the sample size calculated?

Which age groups of participants were included in this study?

What drugs were used for reperfusion therapy? What doses of these drugs were prescribed?

 Results.

The style of English should be improved. The authors did not specify the units of measurement. For example, age - years, score on scales - points, etc.

The median age of the participants is presented without the 25 and 75 quartile.

I do not recommend using abbreviations in table names. Add a name to the first column of Table 1.

The authors report that they have formed three comparable groups, but only two groups are characterized in Table 1.

Probably, the authors formed two groups, but the second group was divided into two subgroups (no-RT with absolute contraindications, no-RT with relative contraindications). This is not clearly explained in the Materials and Methods section, which makes it difficult to understand the results of the study in comparable groups/subgroups.

Figure 1 contains grammatical errors.

Add the name of the first column in Table 5.

 Discussion.

The style of English should be improved.

Describe more clearly how the results of your research differ from previously conducted similar studies by other authors.

It is better to move the Limitations to a separate section.

 Conclusion.

This section should be shortened.

Move the debatable questions with links to other authors to the Discussion section.

Links are not allowed in the Conclusions section.

 References.

 Authors should use the style of references recommended by the journal.

This section needs technical correction.

The quality of English is low. The manuscript is hard to read. I recommend the authors to use the help of the English service of MDPI.

Author Response

Dear Reviewer, 

Thank you for your kind response and time. Please find the response below:

Abstract.

Please note that the purpose here and in the Introduction section should be identical - corrected.

Don't start sentences with numbers - corrected.

Add the mean age of the patients - corrected.

Add the missing spaces between the digits (for example, lines 13, 14, 15 and beyond) - corrected.

 Materials and methods.

Please add a description of the type of this study - corrected.

Add information about the approval of the study by the Ethics Committee, including the institution, registration number and the date of the protocol added.

Formulate clear criteria for inclusion and exclusion of patients in this study - corrected.

Add criteria for forming three groups of participants, including their minimum and maximum age - corrected. Age was not a criterion for inclusion into any particular group. We added the information about the minimal age (18 years old) for the inclusion into the study. 

How was the sample size calculated? - We aimed for study power of at least > 0.8. With the sample collected, study power for the logistic regression was 0.879 (calculated with G-Power). This information has been added.

Which age groups of participants were included in this study? - information added (18 years old and older).

What drugs were used for reperfusion therapy? What doses of these drugs were prescribed? - added (For IVT, 0.9 mg/kg dose of alteplase was used (maximum dose of 90 mg), with 10% of the dose given as a bolus in 1-2 minutes, and the rest given as intravenous infusion for 1 hour)

 Results.

The style of English should be improved. The authors did not specify the units of measurement. For example, age - years, score on scales - points, etc. - reviewed.

The median age of the participants is presented without the 25 and 75 quartile - All of the numeric variables are presented with a single number representing interquartile range. If it is inappropriate, we can change it for all numeric variables.

I do not recommend using abbreviations in table names. Add a name to the first column of Table 1. - corrected.

The authors report that they have formed three comparable groups, but only two groups are characterized in Table 1. Probably, the authors formed two groups, but the second group was divided into two subgroups (no-RT with absolute contraindications, no-RT with relative contraindications). This is not clearly explained in the Materials and Methods section, which makes it difficult to understand the results of the study in comparable groups/subgroups. Thank you for this remark, definitely needed clarification - corrected in the Methods section.

Figure 1 contains grammatical errors - I am so sorry, could you please tell me, where the errors are? If the figure is inappropriate, it can be deleted at all.

Add the name of the first column in Table 5. - added. 

 Discussion.

The style of English should be improved. - corrected.

Describe more clearly how the results of your research differ from previously conducted similar studies by other authors. - corrected.

It is better to move the Limitations to a separate section - corrected

 Conclusion.

This section should be shortened. Move the debatable questions with links to other authors to the Discussion section. Links are not allowed in the Conclusions section - corrected.

Reviewer 2 Report

Firstly, I would like to congratulate the authors on their relevant and interesting work.
However, some points need to be addressed:
1) I advise the authors to work with English editing services to improve the readability of the text.
2) It is advisable that the authors include the study design type in the title.
3) In the abstract, it is important that the authors clarify the study question and goals, and briefly mention the study design type and the year in which the study was conducted.
Introduction
4) It is advisable that the authors improve the introduction by including more information about the pathophysiology of PCS, the most commonly affected arteries, epidemiological data, and comparison of incidence in their country with other nations in the globe.
5) In the introduction and discussion the authors should include more details about the management of PCA, how it differs from the management of anterior circulation strokes, and alternative modern therapies.
6) There are several published retrospective studies about thrombolysis in PCS. The authors should clearly state their research question and highlight what is original about their own study that will add value to the literature.
Methods
7) It is advisable that the authors provide more details about the population seen by their hospital.
8) Please provide IRB number.
9) The authors should explain why the study includes data from 3 years ago and not more recent data. An analysis of the impact of COVID on stroke treatment in their hospital would add value to the study.
10) The authors should provide more details about this study protocol. Is it original or was it created by the authors?
11) Was the study protocol previously validated?
12) Was the study protocol registered?
13) Please provide an accurate description of enrollment methods and inclusion and exclusion criteria
14) Please provide a flow chart describing the selection and exclusion of patients
15) The authors should specify by which professionals were the patients seen (neurologists, neurointerventional specialists)
16) The authors should be objective about the arrival times of the patients, and include number. The term ``belated`` arrival is too subjective and should be excluded.
17) History of stroke should have been evaluated separately from TIA.
18) What does ‘early ambulatory outcome’’ mean?
19) What does ‘lethal outcome’ mean? When were these outcomes analyzed?
20) The hospital protocol for stroke management should be provided.
21) The authors should specify why they chose to analyze these specific parameters and why other relevant parameters were left out, such as smoking, dyslipidemia, stroke etiologies (dissection, embolism, atherosclerosis), types of admission (self initiated, paramedics, emergency service).

Extensive English editing required.

Author Response

Dear Reviewer,

Thank you for your time and useful remarks. Please find the response below.

1) I advise the authors to work with English editing services to improve the readability of the text.
2) It is advisable that the authors include the study design type in the title - corrected.
3) In the abstract, it is important that the authors clarify the study question and goals, and briefly mention the study design type and the year in which the study was conducted. - corrected. I am afraid, though, that we are limited by the amount of the signs allowed in the abstract.

Introduction
4) It is advisable that the authors improve the introduction by including more information about the pathophysiology of PCS, the most commonly affected arteries, epidemiological data, and comparison of incidence in their country with other nations in the globe. - information about the PCS peculiarities was added. 

5) In the introduction and discussion the authors should include more details about the management of PCA, how it differs from the management of anterior circulation strokes, and alternative modern therapies. - added.
6) There are several published retrospective studies about thrombolysis in PCS. The authors should clearly state their research question and highlight what is original about their own study that will add value to the literature. - added.

Methods

7) It is advisable that the authors provide more details about the population seen by their hospital. - added.

8) Please provide IRB number. - added.

9) The authors should explain why the study includes data from 3 years ago and not more recent data. An analysis of the impact of COVID on stroke treatment in their hospital would add value to the study. - corrected. We did not continue observing the patients during COVID-19 pandemic, because it could result in falsely longer times to treatment and arrival. This would make the data inappropriate to use after the pandemic.

10) The authors should provide more details about this study protocol. Is it original or was it created by the authors? 11) Was the study protocol previously validated?12) Was the study protocol registered? - Thank you for this remark, it was unclear. We tried to make it more understandable. This is a retrospective study approved by the Regional and National Bioethics Comittees. Absolute and relative contraindications were distinguished according to standard hospital protocol - we have clarified this (Table 1).  

13) Please provide an accurate description of enrollment methods and inclusion and exclusion criteria - provided.

14) Please provide a flow chart describing the selection and exclusion of patients - apologies for this, we do not have all number required for the flowchart (e.g., the number of the subjects who were transferred to other hospitals). We summarised the strategy in additional paragraph.

15) The authors should specify by which professionals were the patients seen (neurologists, neurointerventional specialists) - corrected

16) The authors should be objective about the arrival times of the patients, and include number. The term ``belated`` arrival is too subjective and should be excluded - the criteria for the late arrival are now described in detail in the Methods section and in Table 1.

17) History of stroke should have been evaluated separately from TIA. - We were concerned about this, too. However, not all of the TIA patients had an MRI fast enough after the TIA. So, it was wuite hard to tell, whether anybody did have a stroke. Another argument in favour of analysing them together would be the fact that the profile of the risk factors and recommended prevention were the same, as for stroke.   

18) What does ‘early ambulatory outcome’’ mean? added
19) What does ‘lethal outcome’ mean? When were these outcomes analyzed? added

20) The hospital protocol for stroke management should be provided. - added.

21) The authors should specify why they chose to analyze these specific parameters and why other relevant parameters were left out, such as smoking, dyslipidemia, stroke etiologies (dissection, embolism, atherosclerosis), types of admission (self initiated, paramedics, emergency service). Thank you for this remark. Smoking status and dyslipidaemia were not collected because of the lack of these data. We did not stratify our population according to the aetiology and type of admission, because this was not the aim of the study. However, this is a relevant limitation, and we will mention it in the Limitations section.

Round 2

Reviewer 1 Report

The authors have modified and improved the manuscript, but it still needs correction. Despite the authors' step-by-step responses to my comments that corrections have been made, most of the corrections have not actually been made.

As before, the design of this study is poorly presented, add a figure explaining the formation of groups and subgroups of participants. Please clearly state the inclusion criteria (1, 2, 3, 4, ...) and exclusion criteria (1. 2, 3, 4, ...).

Lines 54, 56, 102, 103, 124 and further - Explain all abbreviations when using them for the first time. Do not use an abbreviation if it is used less than 4 times in the manuscript.

Line 105 - Correctly write IVT (intravenous thrombolysis), not EVA.

Line 102 - Add units of measurement, for example: NIHSS ≤ 5 points, age > 80 years, etc.

Line 121 - Correct: mRS 0-3 points.

Lines 153 – 164, 176 – 173, etc. - Correct stylistic errors and do not start sentences with numbers.

Lines 154, 161, 179, etc. - Add units of measurement (points).

Table 2: Add information that the median age and interquartile range in years are represented. Add the information that the median estimate of stroke severity according to NIHSS and the interquartile range, in points, are presented.

Table 3: Add units of measurement for mRS, NIHSS and patient age.

Figure 1: I recommend deleting it.

Tables 4, 5: Add units of measurement of patients' age (Patients' age > 80 years).

Table 6: Add the name of the first column.

Earlier, I recommended the authors to add a "Limitations" section. However, this section is still presented as an independent one.

The style of the English language needs serious correction.

The style of the English language needs serious correction.

Author Response

As before, the design of this study is poorly presented, add a figure explaining the formation of groups and subgroups of participants. Please clearly state the inclusion criteria (1, 2, 3, 4, ...) and exclusion criteria (1. 2, 3, 4, ...) - corrected

Lines 54, 56, 102, 103, 124 and further - Explain all abbreviations when using them for the first time. Do not use an abbreviation if it is used less than 4 times in the manuscript - corrected

Line 105 - Correctly write IVT (intravenous thrombolysis), not EVA - EVT ("endovascular treatment") has been corrected to MT ("mechanical thrombectomy"), as it is used in the rest of the article

Line 102 - Add units of measurement, for example: NIHSS ≤ 5 points, age > 80 years, etc. - corrected

Line 121 - Correct: mRS 0-3 points. - corrected

Lines 153 – 164, 176 – 173, etc. - Correct stylistic errors and do not start sentences with numbers - corrected

Lines 154, 161, 179, etc. - Add units of measurement (points) - added

Table 2: Add information that the median age and interquartile range in years are represented. Add the information that the median estimate of stroke severity according to NIHSS and the interquartile range, in points, are presented. - added

Table 3: Add units of measurement for mRS, NIHSS and patient age. - added

Figure 1: I recommend deleting it. - deleted

Tables 4, 5: Add units of measurement of patients' age (Patients' age > 80 years) - added

Table 6: Add the name of the first column. - added

Earlier, I recommended the authors to add a "Limitations" section. However, this section is still presented as an independent one. - Thank you for the useful comment. We added the "Strenghts and limitations" section.

English language has been reviewed by the neurologist with special interest in stroke, who works and lives in the UK.

Reviewer 2 Report

The manuscript has improved after the first correction. However, there are still some questions which should be addressed or that were not sufficiently addressed and I repeat them here:

1)      The authors should provide more details about this study protocol. Is it original or was it created by the authors?

2)      Was the study protocol previously validated?

3)      History of stroke should have been evaluated separately from TIA. These groups must be divided and if a history of stroke is not certain that should be a separate category from stroke that was documented. I suggest dividing patients between those who had TIA, those who had stroke, and those who had an uncertain diagnosis of stroke.

4)      The authors should specify why they chose to analyze these specific parameters and why other relevant parameters were left out, such as smoking, dyslipidemia, stroke etiologies (dissection, embolism, atherosclerosis), types of admission (self initiated, paramedics, emergency service). The study protocol should be clearly explained. Why were such parameters chosen? What was the rationale?

5)      What does ‘early ambulatory outcome’’ mean? The authors did not provide a clear explanation of this definition.

6)      What does ‘lethal outcome’ mean? When were these outcomes analyzed? The authors did not provide a clear explanation. The authors did not provide a clear explanation of this definition.

7)      The authors should also have analyzed other types of treatment for PCS, including endovascular treatments. They also mention `` Mechanical thrombectomy was performed within 6 hours since onset, and within 24 hours in case of BAO.`` however the results are not shown.

8)      Important data about stroke management should have been analyzed, such as time of onset of signs and symptoms to admission, door-to-needle, door-to-puncture, recanalization score.

9)      It would be interesting if authors included the most common presenting signs and symptoms individually of the patients.

10)  The articles should work with a professional statistician to improve the statistical analysis description and reporting.

11)  How was normality assessed?

12)  The information in Figure 1 should be transported to a graph and the figure excluded.

13)  Which parameters were included in logistic regression, how were they selected? Which were the dependent variables? Provide a full description of the methods chosen for inclusion of variables and report relevant parameters of the logistic regression in a table. At least the authors should include B, S.E., Wald, df, Sig., Exp (B) besides CI.

14)  The authors should provide information about the outcomes of the treatments, including percentage of specific adverse effects

Discussion & Conclusion

15)  The authors should provide a comparison of their data with other relevant studies published in the literature and highlight what is new and relevant about their study.

16)  The authors should expand the limitation section and include the aforementioned points.

The authors should mention what could be investigated improved in future studies and future directions.

Author Response

1)      The authors should provide more details about this study protocol. Is it original or was it
created by the authors? 
Reply: Thank you so much for the comment. The study protocol is original, designed by the
authors of the manuscript to fill the knowledge gap on reperfusion treatment of posterior
circulation stroke. It is a component of a larger comprehensive study on the characteristics of
posterior circulation stroke.

2)      Was the study protocol previously validated? 

Reply: Thank you so much for the question. The study protocol has not been validated as it is
a retrospective analysis of patients with posterior circulation stroke treated in routine clinical
practice. Patients were selected for reperfusion treatment according to the internationally
accepted selection (inclusion and exclusion) criteria used in clinical practice worldwide. These
criteria are based on extensive international studies and recommendations (ESOC,
AHA/ASA, national acute stroke guidelines), but the choice of specific treatment modalities is
often left to the discretion and judgment of the treating physician. No additional interventions
were used for the purposes of the study. The aim of the study was not to look for new
treatments or other interventions, but to assess what factors determine the choice of
treatment in routine clinical practice. We believe that this knowledge is important to improve
the safe use and accessibility of reperfusion therapy in a significant proportion of acute stroke
patients who have experienced posterior circulation stroke.

3)      History of stroke should have been evaluated separately from TIA. These groups must be
divided and if a history of stroke is not certain that should be a separate category from stroke that
was documented. I suggest dividing patients between those who had TIA, those who had stroke,
and those who had an uncertain diagnosis of stroke. 

Reply: Thank you so much for the comment. We agree that stroke and TIA as baseline
anamnestic characteristics of patients could be separated into separate rows or groups in
Tables 2, 4 and 5 because stroke and TIA are quantitatively different in terms of the structural
damage to brain tissue. However, the study is a retrospective evaluation of patients treated in
routine practice, where it is virtually impossible to reliably distinguish TIA from stroke without
multimodal neuroimaging studies. Furthermore, even DWI-negative MRI cases (which would
indicate the absence of brain tissue damage) can present with symptoms that last < 24 hours
and are attributed to stroke, while transient symptoms lasting < 24 hours, historically attributed
to TIA, can be accompanied by DWI-positive changes that confirm a stroke (cerebral
infarction). The incorporation of both clinical aspects (i.e., specific syndrome and duration of
symptoms) and the imaging status (i.e., imaging evidence of tissue injury or not) underscores
the importance of considering TIA as part of a continuum rather than a discrete and separate
entity from cerebral infarction (Ortiz-Garcia J et al. Fac Rev 2022;11:19. doi: 10.12703/r/11-
19). The paradigm and definitions of TIA have been changing recently, especially with the
development of neuroimaging studies - due to diagnostic uncertainties, the continuum of
acute cerebrovascular syndromes and overlapping features TIA and minor stroke may be viewed in the same way, both in terms of vascular risk and stroke prevention (Mendelson SJ
et al. JAMA 2021;325(11):1088-1098; doi: 10.1001/jama.2020.26867). In our work, we looked at TIA and stroke together in the anamnesis as an objective confirmation of vascular risk, so we did not separate them. If it seems acceptable to put clinical
TIAs (i.e., focal neurological symptoms that lasted less than 24 hours) into the TIA group, we can
do this. However, the relevance of classifying history in such a way is uncertain, especially, taking
into account that the risk factor profile for TIA and stroke is similar.

4)      The authors should specify why they chose to analyze these specific parameters and why
other relevant parameters were left out, such as smoking, dyslipidemia, stroke etiologies
(dissection, embolism, atherosclerosis), types of admission (self initiated, paramedics, emergency
service). The study protocol should be clearly explained. Why were such parameters chosen?
What was the rationale?

Reply: Thank you so much for the important comment. We did not analyse the influence of
smoking status and dyslipidemia on the likelihood of receiving reperfusion therapy, as data on
these factors were often missing from the medical records due to the retrospective design of
the study. We stated this in the section 4.1. Strengths and limitations of the study. Determining
the exact etiology of ischemic stroke (embolism, atherosclerosis, small vessel disease,
unknown causes, in rare cases - dissection, hypercoagulation, vasospasm) in the emergency
department setting, where reperfusion therapy decisions are made, is often difficult and is
unlikely to have a major impact on these decisions in urgent situation. In international
recommendations and selection criteria, it is not indicated to consider the etiological
differences of ischemic stroke. Nevertheless, we added this in the limitations section.
Your comment about the types of admission is very important and definitely deserves a
separate evaluation. However, these data are beyond the scope of this study. The aim of our
study was to assess clinical patient-related factors, not the organisational process itself. We
mentioned this aspect as our limitation as follows: „We also did not stratify our population
according to the type of admission (self-presented or paramedics)“.

5)      What does „early ambulatory outcome’’ mean? The authors did not provide a clear explanation of this definition. 

Reply: Thank you so much for the question and comment. We explained this in the section 2.
Materials and methods: „early ambulatory outcome defined as mRS 0-3 points (able to walk)
on discharge from the stroke centre“ (line 116).

6)      What does ‘lethal outcome’ mean? When were these outcomes analyzed? The authors did
not provide a clear explanation. 

Reply: Thank you so much for the question and comment. We have clarified and added an
explanation in the section 2. Materials and methods: „Outcomes compared were lethal outcome during the hospitalization (labelled as an early lethal outcome)“ (lines 114-116).

7)      The authors should also have analyzed other types of treatment for PCS, including
endovascular treatments. They also mention `` Mechanical thrombectomy was performed within 6
hours since onset, and within 24 hours in case of BAO.`` however the results are not shown. 

Reply: Thank you so much for the comment. Mechanical thrombectomy (or endovascular
treatment - these terms are used interchangeably) is a method of reperfusion therapy, like intravenous thrombolysis. All the subjects who were treated with any reperfusion therapy method, including intravenous thrombolysis, mechanical thrombectomy (or endovascular treatment) or both (so-called bridging therapy), were included in the study, RT group. This is described in the section 2. Materials and methods: „Subjects treated with intravenous thrombolysis (IVT), mechanical thrombectomy (MT) or both methods (bridging therapy (BT) were included in the RT group...“ (lines 91-92). However, profound analysis and comparison
of the results of any reperfusion therapy method are beyond the scope of this study. Instead, the purpose of the study we present is to evaluate the influence of different demographic and clinical factors on receiving or not receiving reperfusion therapy (either modality).

8)      Important data about stroke management should have been analyzed, such as time of
onset of signs and symptoms to admission, door-to-needle, door-to-puncture, recanalization
score. 

Reply: Thank you so much for the important comment and suggestion. These parameters are
very important in the analysis of reperfusion therapy for posterior circulation stroke (PCS), but, as in response to the previous comment, this information is beyond the scope and purpose of this study, since DNT, DPT, and TICI scores cannot be treated as factors influencing the decision to administer reperfusion therapy. We also analyze key performance indicators of reperfusion methods (such as ODT, DNT, DPT, TICI, etc.) as part of our broader PCS study, but we plan to present this in another manuscript.

9)      It would be interesting if authors included the most common presenting signs and symptoms
individually of the patients.

Reply: Thank you so much for the interesting comment and suggestion. Indeed, strokes of
the posterior circulation are characterized by a very wide variety of symptoms and signs, which often complicates the clinical diagnosis of PCS and differentiation from stroke mimics. We have added another figure illustrating the prevalence of individual symptoms.

10)  The articles should work with a professional statistician to improve the statistical analysis
description and reporting.

Reply: Thank you very much for suggestion. When planning the work, we consulted with a
professional specialist in medical statistics and reviewed the manuscript, making the
necessary corrections and additions. We have contacted him repeatedly.

11)  How was normality assessed? 

Reply: Thank you so much for the question. Yes, we performed the Shapiro-Wilk test to
assess normality and added this in Section 2 (lines 132-133): “The normality was assessed
using Shapiro-Wilk test, all qualitative variables were not normally distributed“.

12)  The information in Figure 1 should be transported to a graph and the figure excluded. 

Reply: Thank you very much for suggestion. The information in Figure 1 has been moved to
Table 3 and the figure has been removed.

13)  Which parameters were included in logistic regression, how were they selected? Which were
the dependent variables? Provide a full description of the methods chosen for inclusion of
variables and report relevant parameters of the logistic regression in a table. At least the authors
should include B, S.E., Wald, df, Sig., Exp (B) besides CI.

Reply: Thank you very much for the questions and suggestions. We used univariate logistic
regression, so every factor was checked. The selection of the variables probably would be
more relevant for the multivariable regression model. The dependent variables are mentioned in the names of the tables: it is belated arrival in Table 4 and getting reperfusion therapy in Table 5. We added B, Standard Error and p-value (Sig.) to OR (Exp B) and CI that were given previously. 

14)  The authors should provide information about the outcomes of the treatments, including
percentage of specific adverse effects.

Reply: Thank you for suggestion. We fully agree that the analysis of outcomes and adverse
events of reperfusion therapy (RT) is particularly important for improving the safe
implementation and use of RT in clinical practice. However, this information is beyond the
scope of the article, as we did not seek to analyse treatment outcome here. The only
subgroup where the outcomes were analysed, was the RCI group - complication rates are
provided in Table 6. We can omit these findings, as they are not directly related to the aim of
the study.

Discussion & Conclusion

15)  The authors should provide a comparison of their data with other relevant studies published
in the literature and highlight what is new and relevant about their study.

Reply: Thank you so much for the suggestion. We have supplemented the discussion section
with comparisons with the results of other authors.

16)  The authors should expand the limitation section and include the aforementioned points. 

Reply: Thank you for suggestion. We have expanded the limitations section according to your
suggestions.

The authors should mention what could be investigated improved in future studies and future
directions.

Reply: Thank you for suggestion. We have supplemented the manuscript with perspectives
and suggestions for further research.